# Benchmarking the Combinatorial Generalizability of Complex Query Answering on Knowledge Graphs

**Zihao Wang**[*]
Department of CSE
HKUST
zwanggc@cse.ust.hk

**Hang Yin**[*]
Department of Mathematical Sciences
Tsinghua University
h-yin20@mails.tsinghua.edu.cn

**Yangqiu Song**
Department of CSE, HKUST
Peng Cheng Laboratory, Shenzhen, China
yqsong@cse.ust.hk

## Abstract

Complex Query Answering (CQA) is an important reasoning task on knowledge graphs. Current CQA learning models have been shown to be able to generalize from atomic operators to more complex formulas, which can be regarded as the combinatorial generalizability. In this paper, we present EFO-1-QA, a new dataset to benchmark the combinatorial generalizability of CQA models by including 301 different queries types, which is 20 times larger than existing datasets. Besides, our benchmark, for the first time, provide a benchmark to evaluate and analyze the impact of different operators and normal forms by using (a) 7 choices of the operator systems and (b) 9 forms of complex queries. Specifically, we provide the detailed study of the combinatorial generalizability of two commonly used operators, i.e., projection and intersection, and justify the impact of the forms of queries given the canonical choice of operators. Our code and data can provide an effective pipeline to benchmark CQA models. [2]

## 1 Introduction

Knowledge graphs, such as Freebase [3], Yago [18], DBPedia [2], and NELL [5] are graph-structured knowledge bases that can facilitate many fundamental AI-related tasks such as reasoning, question answering, and information retrieval [9]. Different from traditional well-defined ontologies, knowledge graphs often have the Open World Assumption (OWA), where the knowledge can be incomplete to support sound reasoning. On the other hand, the graph-structured data naturally provide solutions to higher-order queries such as "*the population of the largest city in Ohio State.*"

Given the OWA and scales of existing knowledge graphs, traditional ways of answering muti-hop queries can be difficult and time-consuming [15]. Recently, several studies use learning algorithms to reason over the vector space to answer logical queries of complex types, e.g., queries with multiple projections [7], Existential Positive First-Order (EPFO) queries [10, 16, 1], and the so called first order queries, i.e., EPFO queries with the negation operator [15, 19, 13]. These tasks are usually called Complex Query Answering (CQA). Unlike the traditional link predictors that only model entities and relations [4], CQA models also consider logical connectives (operators) such as conjunction ($\wedge$),

---

[*]Equal Contribution

[2]https://github.com/HKUST-KnowComp/EFO-1-QA-benchmark

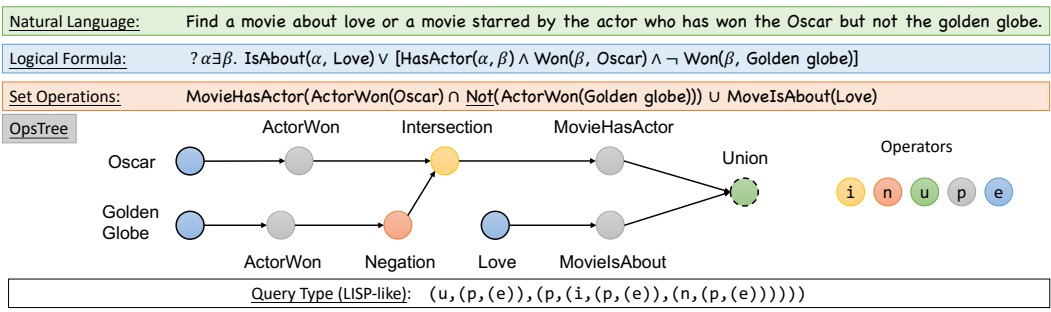

Figure 1: An example of EFO-1 query. The same query are represented in natural language (which is only used for interpreting the query and we do not consider semantic parsing from natural language to logical forms), first-order logical formula, set operations, and OpsTree. The query type can be represented in the LISP-like grammar.

disjunction ($\lor$), and negation ($\neg$) by parameterized operations [10, 16, 15] or non-parameterized operations such as logical t-norms [1, 13].

One of the advantages of learning based methods is that the learned embeddings and parameterization in the vector space can generalize queries from atomic operations to more complex queries. It has been observed there are *out-of-distribution generalization* phenomena of learning models [1] on the Q2B dataset [16] (5 types to train but 4 unseen types to generalize) and the BetaE dataset [15] (10 types to train and 4 unseen types to generalize). This can be explained by the fact that complex queries are all composed by atomic operations such as projection, conjunction, disjunction, and negation. This idea evokes the *combinatorial generalization*, that is, the model generalizes to novel combinations of already familiar elements [21]. However, compared to the huge combinatorial space of the complex queries (see Section 2 and 3), existing datasets [16, 15] only contains queries from very few types, which might be insufficient for the investigation of the combinatorial generalization ability of learning models. Moreover, there is no agreement about how to present the complex queries by operators and normal forms. For example, some approaches treat the negation as the atomic operation [15, 13] while others replace the negation by the set difference (intersection combined with negation) [19, 12]. The impact of the representation of the complex query using learning algorithms is also unclear.

In this paper, we aim to benchmark the combinatorial generalizability of learning models for the CQA on knowledge graphs. We extend the scope from a few hand-crafted query types to the family of Existential First-Order queries with Single Free Variable (EFO-1) (see Section 2) by providing a complete framework from the dataset construction to the model training and evaluation. Based on our framework, the combinatorial generalizability of CQA models that fully supports EFO-1 queries [13, 15, 12] are evaluated and discussed. Our contribution are in three-fold.

- **Large-scale dataset of combinatorial queries.** We present the **EFO-1-QA** dataset to benchmark the combinatorial generalizability of CQA models. EFO-1-QA largely extends the scope of previous datasets by including 301 query types, which is 20 times larger than existing datasets. The evaluation results over three knowledge graphs show that the our set is generally harder than existing ones.

- **Extendable framework.** We present a general framework for (1) iterating through the combinatorial space of EFO-1 query types, (2) converting queries to various normal forms with related operators, (3) sampling queries and their answer sets, and (4) training the CQA models and evaluating the CQA checkpoints. Our framework can be applied to generate new data as well as train and evaluate the models.

- **New findings for normal forms, training, and generalization.** In our dataset, each query is transformed into at most 9 different forms that are related to 7 choices of operators. Therefore, for the first time, a deep analysis of normal forms are available in our benchmark. How the normal form affects the combinatorial generalization is discussed and new observations are revealed. Moreover, we also explore how training query types affect the generalization. We find that increasing training

Table 1: EFO-1 formula for 14 query types in BetaE dataset. The grammar of the EFO-1 formula are given in the Appendix B.

| BetaE | EFO-1 formula | BetaE | EFO-1 formula |
|---|---|---|---|
| 1p | (p,(e)) | 3in | (i,(p,(e)),(i,(p,(e)),(n,(p,(e))))) |
| 2p | (p,(p,(e))) | inp | (p,(i,(p,(e)),(n,(p,(e))))) |
| 3p | (p,(p,(p,(e)))) | pin | (i,(p,(p,(e))),(n,(p,(e)))) |
| 2i | (i,(p,(e)),(p,(e))) | pni | (i,(n,(p,(p,(e)))),(p,(e))) |
| 3i | (i,(i,(p,(e)),(p,(e))),(p,(e))) | 2u-DNF | (u,(p,(e)),(p,(e))) |
| ip | (p,(i,(p,(e)),(p,(e)))) | up-DNF | (u,(p,(p,(e))),(p,(p,(e)))) |
| pi | (i,(p,(p,(e))),(p,(e))) | 2u-DM | (n,(i,(n,(p,(e))),(n,(p,(e))))) |
| 2in | (i,(p,(e)),(n,(p,(e)))) | up-DM | (p,(n,(i,(n,(p,(e))),(n,(p,(e)))))) |

query types is not always beneficial for CQA tasks, which leads to another open problem about how to train the CQA models.

# 2 Complex Queries on KG

In this section, we introduce the Existential First Order Queries with Single Free Variable (EFO-1) on the knowledge graphs. Here we give an intuitive example of EFO-1 queries and the related concepts in Figure 1. Compared to the query families considered in the existing works [10, 16, 15], EFO-1 is a family of queries that are most general. The formal definition and self-contained formal derivation of EFO-1 query family from first-order queries can be found in the Appendix A. Notably, the formal derivation of EFO-1 queries enables and guarantees the logical equivalent query representation in set operations and Operators Tree (OpsTree). Specifically, the formally derived OpsTree is the composition of set functions including intersection, union, negation, projection, and entity anchors. This presentation is also widely but informally introduced in existing CQA models [16, 15, 19].

We consider the EFO-1 queries at the *abstract level* and the *grounded level*. At the abstract level, the structure of a query is specified, but the projections or the entities are not given. At the grounded level, the projections and entities are instantiated (see Section 3 for how to ground the queries). We call queries without the instantiation *query types*. When the query type is given, one can ground the projections and entities in a KG to obtain the specific EFO-1 query.

# 3 The Construction of EFO-1-QA Benchmark

In this section, we cover the detailed framework of the construction of EFO-1-QA benchmark. Our framework includes (1) the generation and normalization of EFO-1 query types following the definitions in Section 2; (2) grounding query types to specific knowledge graph to get the queries and sampling the answer set; (3) constructing the computational graph to conduct the end-to-end training and evaluation. (4) evaluation of models with metrics that emphasize the generalizability. In our practice, we keep the EFO-1-QA dataset as practical as possible and follow the common practice of BetaE dataset. Our benchmark contains 301 different query types (in the Original form) and is at least 20 times larger than the previous works [16, 15, 12]. Moreover, the overall dataset construction and inference pipeline is general enough. It can be applied to EFO-1 queries of any complexity and any properly parametrized operators.

## 3.1 Generation of EFO-1 Query Types

Since EFO-1 queries can be represented by the OpsTree, we employ a LISP-like language [14] to describe the *EFO-1 query types*. The string generated by our grammar is called an *EFO-1 formula* (see Appendix B for more details about the grammar). Follow our derivation of EFO-1 queries from the FO queries (see Appendix A), five operators are naturally introduced by the Skolemization process, including entity e (zero operand), projection p (one operand), negation n (one operand), intersection i (two operands), and union u (two operands). Specifically, Table 1 gives the example of

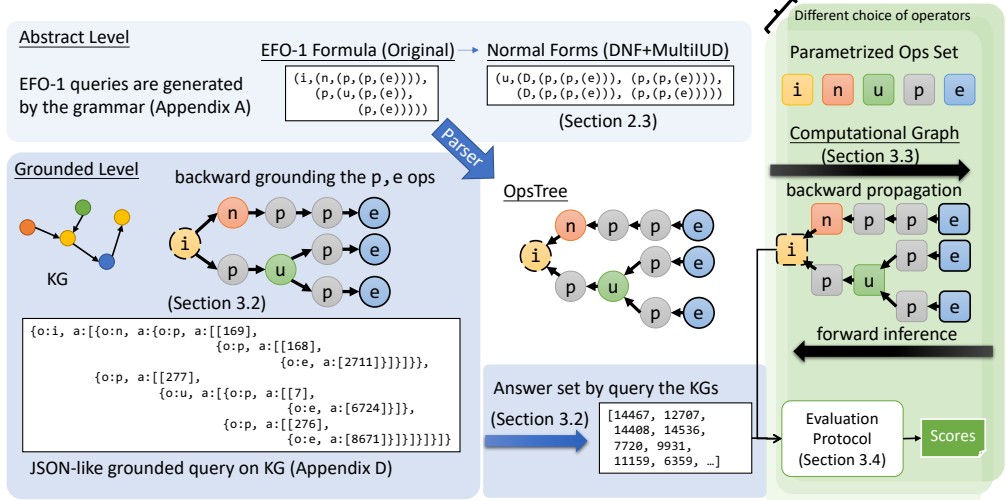

Figure 2: Framework of constructing the EFO-1-QA benchmark. The query types are defined by the EFO-1 formula, which is generated by the Grammar 2 in the Appendix B at the abstract level. The EFO-1 formula can be converted to different normal forms and represented with the different operators. A parser is employed to produce the OpsTree from the EFO-1 formula. Queries are grounded by the backward DFS of the OpsTree on the full graph and the answers are sampled by the forward execution of the OpsTree on the partial graph as explained in Section 3.3. The OpsTree can also be used to build the computational graph with the parameterized operators, which are used to train and infer the CQA models by the backward propagation and forward inference. Finally, the estimated query embeddings are evaluated by the Evaluation Protocol with five metrics given in the Section 3.5.

Table 2: Number of EFO-1 query types with respect to the maximum length of projection chains and number of anchor nodes for EFO-1-QA and BetaE dataset. The boldface indicates the query types that are not discussed sufficiently in the BetaE dataset.

| Max length of projection chains | # Anchor nodes | | | | | | | |
| | EFO-1-QA | | | | BetaE | | | |
| | 1 | 2 | 3 | Sum | 1 | 2 | 3 | Sum |
| --- | --- | --- | --- | --- | --- | --- | --- | --- |
| 1 | 1 | 3 | 12 | 16 | 1 | 3 | **2** | 6 |
| 2 | 1 | 10 | 91 | 102 | 1 | **6** | **0** | 7 |
| 3 | 1 | 13 | 169 | 183 | 1 | **0** | **0** | 1 |
| Sum | 3 | 26 | 272 | 301 | 3 | 9 | 2 | 14 |

the EFO-1 formulas for 14 query types in the BetaE dataset [15]. We can parse any EFO-1 formula to the OpsTree according to our grammar.

In EFO-1-QA benchmark, the EFO-1 formulas are generated by a depth first search of the Grammar 2 in the Appendix B with the [e,p,i,u,n] operators. The grammar explicitly follows the practice of *bounded negation*. That is, we only generate the negation operator when it is one operand of an intersection operator. The produced OpsTree is binary.

Instead of producing endless query types in the combinatorial spaces of EFO-1 queries, we keep the generated types as realistic as possible by following two practical constraints: (1), we set the maximum length of projection/negation chains to be 3. That is, we consider no more than three projections/negations in any paths from the target root node to anchor leaf nodes, which follows the 3p setting in Table 1. (2), we limit the number of anchor nodes to be no more than 3, which follows the 3i setting in Table 1. As a result, we generate 301 different query types, more details can be found in Table 2.

Table 3: The normal forms of logical queries, related choice of operators and the number of types of each normal form considered in EFO-1-QA benchmark.

| (Normal) Forms | Operators | Comment |
|---|---|---|
| Original | [e,p,i,u,n] | Sort multiple operands by the alphabetical order |
| DM | [e,p,i,n] | Replace the u with i,n by De Morgan's rule |
| DM + I | [e,p,I,n] | Replace i in DM by I |
| Original + d | [e,p,i,u,d] | Replace i-n structure by binary d operator |
| DNF | [e,p,i,u,n] | Disjunctive Normal Form derived by the Appendix C |
| DNF + d | [e,p,i,u,d] | Replace the n in DNF by binary d |
| DNF + IU | [e,p,I,U,n] | Replace the binary i,u in DNF by I,U |
| DNF + IUd | [e,p,I,U,d] | Replace n in DNF+IU by binary d |
| DNF + IUD | [e,p,I,U,D] | Replace the n in DNF+IU by multi-difference D |

## 3.2 Normalization of EFO-1 Query Types

Interestingly, in the context of learning based CQA models, the logically equivalent transformation of query types may lead to computationally different structures. On the one hand, different choices of operators lead to different parameterizations and generalization performances. For example, the set difference operator [12] is reported to perform differently from the negation operator [15]. On the other hand, different normal forms also affect the learning based CQA models. Specifically, different forms alters the query structure, i.e., OpsTree, and might result in different depths or various number of inputs of the specific operator (see the DNF formula and the DNF+IU formula in the Appendix D Table 12) and finally affect the performance. For example, DNF has been claimed to be better than the De Morgan by [15] when evaluating on 2u and up queries.

However, the impacts of the operators and normal forms are not clearly justified in previous works because they are also entangled with parametrization, optimization, and other issues. Our benchmark, to the best of our knowledge, is the first to justify the impact of operators and normal forms from the aspect of the dataset. Our LISP-like language is general enough to be compatible with all those different query types. Here we list how EFO-1-QA benchmark considers the impact of choices of operators (see the Grammar 3 in the Appendix B) and normal forms.

**(A) Choice of the Operators.** We have introduced the [e,p,i,u,n] operator system by Skolemization. In BetaE dataset [15], multi-intersection operator I and multi-union operator U that accept more than two inputs to conduct the intersection and union are chosen in the [e,p,I,U,n] system. In this case, the "3i" type in Table 1 can be rewritten as (I,(p,(e)),(p,(e)),(p,(e))). Moreover, the set difference operator d or the generalized multi-difference operator D are introduced in [12] to replace the negation operator n for EFO-1 queries with the *bounded negation* assumption. The rationale behind the *bounded negation* is that the negation should be bounded by a set intersection operation because the set complement against all entities is not practically useful. So one can replace each intersection-negation structure with the set difference, resulting in [e,p,i,u,d] or [e,p,I,U,D] systems. However, the removal of the negation operator made it impossible to apply the De Morgan's law, which can represent the union operator u by intersection i and negation n. More comment of the operators can also be found in the Appendix B. To summarize, we consider seven choices of operators to represent the EFO-1 queries, see Table 3.

**(B) Choice of Normal Forms.** Normal forms, such as Disjunctive Normal Forms (DNF) [8], are equivalent classes of query types. Normalization, i.e., converting queries into normal forms, is effective to reduce the number of query types and rectify the estimation process while preserving the logical equivalence. The participation of different operator systems makes the choices of normal forms more complicated. In this work, all nine different forms with seven different choices of operators are shown in Table 3. This nine normal forms are selected by enumerating all possible combinations of operators, see Appendix H. The example of each form and how they are transformed are shown in Table 12 in the Appendix D. After obtaining a query from the generation procedure, we transform them to DNF and other seven forms. Most of the conversions are straightforward except the conversion from the original form to the DNF.

### 3.3 Grounding EFO-1 Queries and Sampling the Answer Sets

Given the specific knowledge graph, we can ground the query types with the containing relations and entities. We consider the knowledge graph $\mathcal{G}$ and its training subgraph $\mathcal{G}_{train}$, such that $\mathcal{G}_{train} \subset \mathcal{G}$. To emphasize on the generalizability of CQA models that are trained on $\mathcal{G}_{train}$, the queries are grounded to the entire graph $\mathcal{G}$ and we pick the answers that can be obtained on the $\mathcal{G}$ but not the $\mathcal{G}_{train}$. We note that this procedure follows the protocol in [16] and prevents the data leakage.

**Grounding Query Types.** The grounding means to assign specific relations and entities from the $\mathcal{G}$ to the p and e operators in the OpsTree. We conduct the grounding process in the *reverse* order, i.e., from the target root node to the leaf anchor nodes, as shown in Figure 2. We first sample an entity as the seed answer at the root node and go through the tree. During the iterating, the inputs of each operator are derived by its output. For the set operators such as intersection, union, and negation, we select the inputs sets while guaranteeing the output. For the projection operator, we sample the relation from the reverse edges in the $\mathcal{G}$ that leaves the specific output entity. For the entity operator, i.e., the anchor nodes, we sample the head entity given the relation and the tail entity. In this way, we ensure grounded queries to have at least one answer. The sampling procedure for the negation operator is actually a bit more complicated and we leave the details in the Appendix F. In order to store the grounded relation and query information, we employ the JSON format to serialize the information. The details of the JSON string can be found in the Appendix E.

**Sampling Answer Sets.** Once the query is grounded, we can sample the answer by the execution of the OpsTree in the full knowledge graph $\mathcal{G}$. The execution procedure of each operator is defined in the Table 11. The full answer set $A_{full}$ is obtained on the $\mathcal{G}$ and the trivial answer set $A_{trivial}$ is obtained by sampling the training subgraph $\mathcal{G}_{train}$. As we stated, we focus on the answer set $A = A_{full} - A_{trivial}$ that cannot be obtained by simply memorizing the known training graph $\mathcal{G}_{train}$. Specifically, we pick the queries whose answer sizes are between 1 and 100, which follows the practice of BetaE dataset [15].

For each query type, we can produce one data sample by a grounding and sampling process. We note that the grounding and sampling process does not rely on a specific graph. In this work, we sample the benchmark dataset on three knowledge graphs, including FB15k [20], FB15k-237 [4], and NELL [5] with 5000, 8000, and 4000 queries correspondingly. More details about how the dataset is organized can be found in the Appendix G.

### 3.4 From OpsTree to Computational Graph

Similar to the sampling process where the answers are drawn by the forward computation of the OpsTree, we can also construct the end-to-end computational graph with the parameterized operators in the same topology to estimate the answer embeddings. Therefore, we can train and evaluate the CQA models over the constructed computational graphs of all EFO-1 queries. Practically, we can even use any provided checkpoints to initialize the parameterized operators and conduct the inference. Therefore, the EFO-1-QA provides a general test framework of CQA checkpoints with no need to know how the checkpoints are obtained.

### 3.5 Evaluation Protocol

The CQA models are evaluated by the ranking based metrics in the EFO-1-QA benchmark. Basically, the ranking of all entities are expected to be obtained after the inference. For example, the entities can be ranked by their "distances" to the estimated answer embedding. We use following metrics to evaluate the generalizability of CQA models, including MRR and HIT@K that have been widely used in previous works [15, 1, 12].

• **MRR.** For each answer entity in the answer set, we consider its ranking with $\mathcal{E} - A_{full}$.[3] That is, the ranking of the given answer against all non-answer entities. The Mean Reciprocal Rank (MRR) for a query is the average of the MRR of all answers of this query. The MRR of a query can be 1 if all the answers are ranked before the rest non-answer entities. Then the query MRR are averaged to the specific query types or the entire dataset.

---

[3]In query with negation, this should be $\mathcal{E} - A_{full} - A_{trivial}$ instead. The rationale behind it is that $A_{trivial} \not\subset A_{full}$ for negation creates "wrong" answers in $A_{trivial}$.

Table 4: Review of existing CQA datasets, where * means the DNF/DM is required.

| CQA Dataset | Support Operators | | | | | | | | | Support EPFO | Support EFO-1 | Num. of Forms | Num. of Test Query Types |
|---|---|---|---|---|---|---|---|---|---|---|---|---|---|
| | e | p | i | I | u | U | n | d | D | | | | |
| Q2B dataset [16] | ✓ | ✓ | ✓ | ✓ | ✗ | ✗ | ✗ | ✗ | ✗ | ✓* | ✗ | 1 | 9 |
| HypE dataset [6] | ✓ | ✓ | ✓ | ✓ | ✗ | ✗ | ✗ | ✗ | ✗ | ✓* | ✗ | 1 | 9 |
| BetaE dataset [15] | ✓ | ✓ | ✓ | ✓ | ✗ | ✗ | ✓ | ✗ | ✗ | ✓* | ✓* | 2 | 14 |
| EFO-1-QA (ours) | ✓ | ✓ | ✓ | ✓ | ✓ | ✓ | ✓ | ✓ | ✓ | ✓ | ✓ | 9 | 301 |

Table 5: Benchmark results of MRR (%) on different dataset. The results of the BetaE dataset are obtained from the original paper [15, 13].

| CQA Model | Dataset | FB15k-237 | | | FB15k | | | NELL | | |
|---|---|---|---|---|---|---|---|---|---|---|
| | | EPFO | Neg. | ALL | EPFO | Neg. | ALL | EPFO | Neg. | ALL |
| BetaE | BetaE | 20.9 | **5.4** | 15.4 | 41.6 | **11.8** | 31.0 | 24.6 | **5.9** | 17.9 |
| +DNF+IU | EFO-1-QA | **11.8** | 7.5 | **9.7** | **23.7** | 16.8 | **20.3** | **12.7** | 8.3 | **10.6** |
| LogicE | BetaE | 22.3 | **5.6** | 16.3 | 44.1 | **12.5** | 32.8 | 28.6 | **6.2** | 20.6 |
| +DNF+IU | EFO-1-QA | **12.8** | 8.1 | **10.5** | **25.4** | 18.2 | **21.9** | **15.6** | 10.4 | **13.1** |

• **HIT@K.** Similar to MRR, HIT@K is computed for each answer by its ranking in $\mathcal{E} - A_{full}$ and then averaged for the query. In our practice, we consider $K = 1, 3, 10$.

• **Retrieval Accuracy (RA).** Previous metrics focus on the answer entity against non-answer entities, which deviates from the real-world retrieval task. In this paper, we propose the RA score to evaluate how well a model retrieves the entire answer set. The computation of RA score is decomposed into two steps, i.e., (1) to estimate the size of the answer set as $N$, (2) to compute the accuracy of the top-$N$ answers against the true answer set.

We note that EFO-1-QA also supports the counting task. However, since not all the CQA models are designed to count the number of answers, we assume that the ground-truth of the answer size is known and only consider the second step of computing the RA score in this paper. We call the RA score with known answer size as the RA-Oracle. Moreover, as this benchmark focuses on the generalization property of CQA models, we do not report the evaluation in the entailment setting [19].

## 4 Related Datasets and the Comparison to EFO-1-QA Benchmark

Existing datasets are constructed along with the CQA models, for the purpose of indicating that their models are capable to solve some certain types of queries by providing a few examples. Thus, those datasets contain very limited query types, normal forms and operators, see Table 4. However, EFO-1-QA benchmark focuses on how well CQA models work on the whole EFO-1 query space and considers the impact of operators and normal forms.

Table 2 already shows that EFO-1-QA benchmark contains much more query types, supported operators and normal forms than BetaE dataset [15], thereby provides a *more comprehensive* evaluation result. Meanwhile, we compare results of both BetaE [15] and LogicE [13] between EFO-1-QA benchmark and BetaE dataset [15] in Table 5. We note that the EFO-1-QA benchmark is *generally harder* than BetaE dataset when averaging results from all query types on three KGs. Moreover, our comprehensive benchmark brings us many new insights and helps us to refresh the observations from previous dataset.

**Finding 1: Negation queries ares not significantly harder.** We further separate the query types into two subgroups, i.e., the EPFO queries and the negation queries. Table 5 shows that results from two dataset have very different distribution of the scores in those two subgroups. This can be explained by the fact that the five negation query types in the BetaE dataset are biased and cannot represent the general performance of the negation queries.

In short, we can conclude that the EFO-1-QA benchmark is more comprehensive, generally harder, and fairer than existing datasets.

Table 6: Benchmark results (%) for all three models and their corresponding normal forms.

| Knowledge Graph | CQA Moddel Normal Form | BetaE | | | LogicE | | | NewLook | |
|---|---|---|---|---|---|---|---|---|---|
| | | DM | DM +I | DNF +IU | DM | DM +I | DNF +IU | DNF +IUd | DNF +IUD |
| FB15k -237 | MRR | 8.48 | 8.50 | 9.67 | 10.00 | 10.01 | **10.46** | 9.11 | 9.13 |
| | HIT@1 | 4.35 | 4.37 | 4.89 | 5.26 | 5.27 | **5.42** | 4.80 | 4.81 |
| | HIT@3 | 8.54 | 8.56 | 9.69 | 10.19 | 10.21 | **10.61** | 9.14 | 9.15 |
| | HIT@10 | 16.25 | 16.27 | 18.73 | 19.04 | 19.06 | **20.01** | 17.17 | 17.20 |
| | RA-Oracle | 11.49 | 11.51 | 13.69 | 13.63 | 13.65 | **14.37** | 12.43 | 12.45 |
| FB15k | MRR | 17.18 | 17.22 | 20.31 | 20.53 | 20.55 | **21.89** | 19.80 | 19.87 |
| | HIT@1 | 10.46 | 10.51 | 12.05 | 12.68 | 12.70 | **13.14** | 11.96 | 11.99 |
| | HIT@3 | 18.76 | 18.81 | 22.10 | 22.71 | 22.73 | **24.17** | 21.58 | 21.66 |
| | HIT@10 | 30.30 | 30.35 | 36.74 | 35.93 | 35.96 | **39.33** | 35.28 | 35.44 |
| | RA-Oracle | 21.83 | 21.89 | 27.51 | 26.92 | 26.95 | **29.38** | 26.57 | 26.66 |
| NELL | MRR | 8.93 | 8.94 | 10.58 | 11.13 | 11.14 | **13.07** | 9.88 | 9.90 |
| | HIT@1 | 5.58 | 5.59 | 6.52 | 7.26 | 7.27 | **8.31** | 6.04 | 6.04 |
| | HIT@3 | 9.38 | 9.39 | 11.12 | 11.89 | 11.89 | **14.01** | 10.35 | 10.36 |
| | HIT@10 | 15.27 | 15.29 | 18.32 | 18.38 | 18.39 | **22.04** | 17.10 | 17.13 |
| | RA-Oracle | 12.08 | 12.09 | 14.98 | 15.25 | 15.26 | **18.39** | 14.15 | 14.16 |

## 5 The Empirical Evaluation of the Benchmark

In this section, we present the evaluation results of the complex query answering models that are compatible to the EFO-1 queries.

### 5.1 Complex Query Answering Models

We summarize existing CQA models by their supported operators as well as supported query families in Table 13. Only three CQA models fully support EFO-1 family by their original implementation. Therefore, in our evaluation, we focus on these models, including BetaE [15], LogicE [13], and NewLook [12]. These models are trained on the BetaE training set and evaluated on EFO-1-QA benchmark. Specifically, the BetaE is trained by the original implementation released by the authors [4] and evaluated in our framework. LogicE and NewLook are re-implemented, trained and tested by our framework. The NewLook implementation is adapted to fit into the generalization evaluation, see the Appendix I.

### 5.2 Benchmark Results

The benchmark result is shown in Table 6 for three models with five supported normal forms in total on three KGs. Besides the findings in Table 5, the average HIT@1 of NewLook is reported to be 37.0 in their paper [12] but is 4.8 on our EFO-1-QA. This can be caused by the hardness of our dataset and our implementation prevent the data leakage. We also group the 301 query types into 9 groups by their depth and width. The detailed results of FB15K-237 can be found in Table 7. For FB15K and NELL, the corresponding results are listed in the Table 15 and Table 16 in the Appendix L. Detailed analysis in the Appendix L justifies the impact of query structures, for the first time.

## 6 Analysis of the [e,p,i,u,n] System

As discussed in Section 3.1, a CQA model may model queries with multiple choices of operators, which are different in computing while equivalent in logic. We here focus on the canonical choice of [e,p,i,u,n] since this system is naturally derived by Skolemization, represents EFO-1 queries without any assumptions such as *bounded negation*. The best model LogicE in Table 6 is picked in this section.

---

[4] https://github.com/snap-stanford/KGReasoning

Table 7: Benchmark results(%) on FB15k-237. The mark † indicates the query groups that previous datasets have not fully covered. The boldface indicates the best scores. The best scores of the same model are underlined.

| CQA Model | Normal Form | Metric | Query type groups (# anchor nodes, max length of Projection chains) | | | | | | | | | AVG. |
|---|---|---|---|---|---|---|---|---|---|---|---|---|
| | | | (1,1) | (1,2) | (1,3) | (2,1) | (2,2)† | (2,3)† | (3,1)† | (3,2)† | (3,3)† | |
| BetaE | DM | MRR | 18.79 | 9.72 | 9.64 | 12.76 | 8.48 | 8.10 | 11.34 | 8.58 | 8.09 | 8.48 |
| | | HIT@1 | 10.63 | 4.63 | 4.68 | 7.07 | 4.13 | 3.89 | 5.99 | 4.42 | 4.16 | 4.35 |
| | | HIT@3 | 20.37 | 9.61 | 9.44 | 13.47 | 8.37 | 8.02 | 11.99 | 8.66 | 8.11 | 8.54 |
| | | HIT@10 | 36.19 | 19.80 | 19.38 | 24.27 | 16.82 | 16.03 | 21.99 | 16.41 | 15.43 | 16.25 |
| | | RA-Oracle | 14.38 | 14.40 | 16.99 | 14.09 | 12.07 | 13.04 | 12.51 | 10.86 | 11.48 | 11.49 |
| | DM +I | MRR | 18.79 | 9.72 | 9.64 | 12.76 | 8.48 | 8.10 | 11.39 | 8.59 | 8.12 | 8.50 |
| | | HIT@1 | 10.63 | 4.63 | 4.68 | 7.07 | 4.13 | 3.89 | 6.05 | 4.43 | 4.19 | 4.37 |
| | | HIT@3 | 20.37 | 9.61 | 9.44 | 13.47 | 8.37 | 8.02 | 12.01 | 8.68 | 8.14 | 8.56 |
| | | HIT@10 | 36.19 | 19.80 | 19.38 | 24.27 | 16.82 | 16.03 | 22.01 | 16.43 | 15.47 | 16.27 |
| | | RA-Oracle | 14.38 | 14.40 | 16.99 | 14.09 | 12.07 | 13.04 | 12.58 | 10.88 | 11.52 | 11.51 |
| | DNF +IU | MRR | 18.79 | 9.72 | 9.64 | 14.39 | 9.28 | 8.86 | 13.14 | 9.76 | 9.32 | 9.67 |
| | | HIT@1 | 10.63 | 4.63 | 4.68 | 7.78 | 4.48 | 4.20 | 6.83 | 4.93 | 4.72 | 4.89 |
| | | HIT@3 | 20.37 | 9.61 | 9.44 | 15.11 | 9.12 | 8.74 | 13.86 | 9.79 | 9.28 | 9.69 |
| | | HIT@10 | 36.19 | 19.80 | 19.38 | 28.04 | 18.55 | 17.67 | 25.82 | 18.90 | 17.95 | 18.73 |
| | | RA-Oracle | 14.38 | 14.40 | 16.99 | 16.87 | 13.58 | 14.69 | 15.39 | 12.93 | 13.83 | 13.69 |
| LogicE | DM | MRR | 20.71 | 10.70 | 10.18 | 15.66 | 10.01 | 9.41 | 13.71 | 10.12 | 9.54 | 10.00 |
| | | HIT@1 | 11.66 | 5.20 | 5.25 | 8.81 | 5.00 | 4.83 | 7.38 | 5.27 | 5.06 | 5.26 |
| | | HIT@3 | 23.02 | 10.66 | 9.96 | 16.72 | 10.07 | 9.43 | 14.57 | 10.33 | 9.67 | 10.19 |
| | | HIT@10 | 39.81 | 21.25 | 19.48 | 29.66 | 19.66 | 18.12 | 26.33 | 19.38 | 18.04 | 19.04 |
| | | RA-Oracle | 15.64 | 15.27 | 17.28 | 17.49 | 13.97 | 14.75 | 15.62 | 12.99 | 13.61 | 13.63 |
| | DM +I | MRR | 20.71 | 10.70 | 10.18 | 15.66 | 10.01 | 9.41 | 13.76 | 10.14 | 9.56 | 10.01 |
| | | HIT@1 | 11.66 | 5.20 | 5.25 | **8.81** | 5.00 | 4.83 | **7.41** | 5.28 | 5.07 | 5.27 |
| | | HIT@3 | 23.02 | 10.66 | 9.96 | 16.72 | 10.07 | 9.43 | 14.67 | 10.35 | 9.69 | 10.21 |
| | | HIT@10 | 39.81 | 21.25 | 19.48 | 29.66 | 19.66 | 18.12 | 26.41 | 19.42 | 18.06 | 19.06 |
| | | RA-Oracle | 15.64 | 15.27 | 17.28 | 17.49 | 13.97 | 14.75 | 15.64 | 13.01 | 13.63 | 13.65 |
| | DNF +IU | MRR | 20.71 | 10.70 | 10.18 | **15.86** | **10.27** | **9.67** | **14.06** | **10.56** | **10.06** | **10.46** |
| | | HIT@1 | 11.66 | 5.20 | 5.25 | 8.69 | **5.06** | **4.87** | 7.36 | **5.41** | **5.27** | **5.42** |
| | | HIT@3 | 23.02 | 10.66 | 9.96 | **16.85** | **10.31** | **9.66** | **14.90** | **10.71** | **10.16** | **10.61** |
| | | HIT@10 | 39.81 | 21.25 | 19.48 | **30.62** | **20.26** | **18.70** | **27.48** | **20.39** | **19.06** | **20.01** |
| | | RA-Oracle | 15.64 | 15.27 | 17.28 | **17.94** | **14.46** | **15.31** | **15.99** | **13.64** | **14.48** | **14.37** |
| NewLook | DNF +IUd | MRR | **22.31** | **11.19** | **10.39** | 16.02 | 9.46 | 9.29 | 11.54 | 8.62 | 8.95 | 9.11 |
| | | HIT@1 | **13.55** | **5.62** | **5.18** | 9.42 | 4.85 | 4.85 | 6.17 | 4.47 | 4.74 | 4.80 |
| | | HIT@3 | **24.62** | **11.40** | **10.38** | 17.31 | 9.44 | 9.19 | 12.03 | 8.62 | 8.93 | 9.14 |
| | | HIT@10 | **40.53** | **22.18** | **20.47** | 29.10 | 18.20 | 17.58 | 22.21 | 16.34 | 16.76 | 17.17 |
| | | RA-Oracle | **17.66** | **16.32** | **17.79** | 17.53 | 13.00 | 14.66 | 12.40 | 10.85 | 12.91 | 12.43 |
| | DNF +IUD | MRR | **22.31** | **11.19** | **10.39** | 16.02 | 9.46 | 9.29 | 11.59 | 8.65 | 8.96 | 9.13 |
| | | HIT@1 | **13.55** | **5.62** | **5.18** | 9.42 | 4.85 | 4.85 | 6.19 | 4.48 | 4.74 | 4.81 |
| | | HIT@3 | **24.62** | **11.40** | **10.38** | 17.31 | 9.44 | 9.19 | 12.06 | 8.65 | 8.94 | 9.15 |
| | | HIT@10 | **40.53** | **22.18** | **20.47** | 29.10 | 18.20 | 17.58 | 22.33 | 16.41 | 16.78 | 17.20 |
| | | RA-Oracle | **17.66** | **16.32** | **17.79** | 17.53 | 13.00 | 14.66 | 12.43 | 10.88 | 12.92 | 12.45 |

## 6.1 Combinatorial Generalizability of Operators

Since the projection operator plays a pivotal role in query answering as shown in Appendix L, For projection, we train models by {1p}, {1p,2p}, and {1p,2p,3p} queries and evaluate on 1p,2p,3p,4p,5p[5]. The experiment result is shown in the Table 8. We can see that training on deeper query types benefits the generalization power as the performances on unseen query types are improved. However, the performance on 1p decreases at the same time.

For the intersection, we train models by {1p,2i} and {1p,2i,3i}[6] queries and evaluate on 2i,3i,4i queries.[7] As shown in Table 9, adding 3i to training queries helps with the performance on 3i,4i while detriments performance on 2i.

**Finding 2: More training query types do not necessarily lead to better performance.** Adding more queries to training is not helpful to all query types, since it may benefit some query types while impairing others. Our observation indicates the interaction mechanisms between query types is not clear. Thus, how to properly train the CQA models is still open.

---

[5] 1p,2p,and 3p are shown in Table 1, 4p and 5p are defined similarly.

[6] 1p is also included in training to ensure the performance the projection.

[7] 2i and 3i are shown in Table 1, 4i is defined as (i,(i,(i,(p,(e)),(p,(e))),(p,(e))),(p,(e))).

Table 8: Generalization performance of projection on FB15k-237 in MRR (%).

| Training | 1p | 2p | 3p | 4p | 5p |
|---|---|---|---|---|---|
| 1p | 19.36 | 4.98 | 3.95 | 3.17 | 2.93 |
| 1p,2p | 19.22 | 9.01 | 7.98 | 7.22 | 7.15 |
| 1p,2p,3p | 17.81 | 9.45 | 9.59 | 9.52 | 9.32 |

Table 9: Generalization performance of intersection on FB15k-237 in MRR (%).

| Training | multi-input I | | | binary input i | | |
|---|---|---|---|---|---|---|
| | 2i | 3i | 4i | 2i | 3i | 4i |
| 1p,2i | 32.24 | 41.66 | 52.37 | 32.24 | 41.66 | 51.78 |
| 1p,2i,3i | 31.97 | 42.67 | 52.70 | 31.97 | 42.32 | 52.10 |

Table 10: Impact of normal forms of LogicE on FB15k-237. Each cell indicates the winning rate of the form by its row against the form by its column.

| Outperform Rate % | Original | DM | DM+I | DNF | DNF+IU |
|---|---|---|---|---|---|
| Original | 0.00 | 85.96 | 60.61 | 53.33 | 43.33 |
| DM | 14.04 | 0.00 | 41.33 | 12.23 | 20.31 |
| DM+I | 39.39 | 58.67 | 0.00 | 28.50 | 11.60 |
| DNF | 46.67 | 87.77 | 71.50 | 0.00 | 41.67 |
| DNF+IU | 56.67 | 79.69 | 88.40 | 58.33 | 0.00 |

**Finding 3: More complex queries do not necessarily have worse performance.** We can see that the more complex p queries are, the worse performance they have. However, for i queries, more complex i/I queries have better performance. In the combinatorial space where those two operators are combined, we cannot even conclude more complex queries have worse performance or not. This might support our observation that negation queries are not significantly harder since negation operator is assumed to be bounded by an intersection operator.

### 6.2 Impact of the Normal Forms

To study the impact of different normal forms, except for the averaged results in Table6 and Table7, we also compares every normal forms with LogicE [13] with our evaluation model and the results are shown in Table 10 and Table 14 in the Appendix K.

• **DM vs. DNF.** Formulas with unions can be modeled in two different ways: (1) transformed into Disjunctive Normal Form (DNF) as showed in Appendix C, (2) with union converted to intersection and negation by the De Morgan's law (DM). In Table 10, we find that DNF outperforms DM in the vast majority of cases, whether DM uses I or not. However, there are still some cases where DM can outperform DNF.

• **Original vs. DNF+IU.** DNF+IU outperforms all other normal forms. Moreover, it is a universal form to support all circumstances, making itself the most favorable form. Interestingly, the original form, meanwhile, has considerable winning rate against DNF, suggesting it has its own advantage in modeling.

**Finding 4: There is no rule of thumb for choosing the best normal form.** When evaluated on BetaE dataset, one may observe that the DNF is always better than DM. However, in EFO-1-QA, our evaluation shows that there is no normal form that can outperform others in every query types. Thus, how to choose the normal form for specific query type to obtain the best inference-time performance is also an open problem.

## 7 Conclusion

In this paper, we present a framework to investigate the combinatorial generalizability of CQA models. With this framework, the EFO-1-QA benchmark dataset is constructed. Comparisons between existing dataset shows that EFO-1-QA data is more comprehensive, generally harder and fairer. The detailed analysis justifies, for the first time, the impact of the choices of different operators and normal forms. Notably, our evaluation leads four insightful findings that refreshes the observations on previous datasets. Two findings also leads to the open problems for training and inference of the CQA models. We hope that our framework, dataset, and findings can facilitate the related research towards combinatorial generalizable CQA models.

## Acknowledgement

The authors of this paper were supported by the NSFC Fund (U20B2053) from the NSFC of China, the RIF (R6020-19 and R6021-20) and the GRF (16211520) from RGC of Hong Kong, the MHKJFS (MHP/001/19) from ITC of Hong Kong.

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
