# OpenReview forum: "Benchmarking the Combinatorial Generalizability of Complex Query Answering on Knowledge Graphs"
_NeurIPS.cc/2021/Track/Datasets_and_Benchmarks/Round2 — NeurIPS 2021 Datasets and Benchmarks Track (Round 2)_

### Official Review · Reviewer_7kZd · 2021-09-18
**A synthetic dataset for evaluating complex query answering methods**

**Rating:** 6
**Confidence:** 3
**Correctness:** I do not see any methodological issue…

**Strengths:**

- The provided dataset fills a gap in combinatorial generalizability of CQA models evaluation
- Both code and data are available (however, the license is not specified)

**Weaknesses:**

- The synthetic nature of the dataset is not justified
- Lack of clarity when describing the dataset and benchmark

**Additional Feedback:**

After the clarifications and updates provided by the authors, I have updated my score.

**Clarity:**

The paper and the supplementary materials in great detail and clarity describe the problem for which the dataset has been created. I believe the paper would benefit from adding the evidence that real-life CGA on KGs benefits from tuning or evaluating combinatorial generalizability datasets.

Since the authors paid great attention to combinatorial generalizability, I expected this paper to focus on the benchmark, analysis, and discoveries rather than the dataset itself. I think this is the main weakness of the present form of the paper.

**Documentation:**

The authors provide the source code and the data. However, neither code nor data have been provided with the licenses under which they are distributed. Although the source code is relatively easy to read, it has no documentation or docstrings besides the well-written README file. The repository also contains blocks of commented code without a proper explanation. I am concerned about the convenience of the current dataset representation in CSV, as it requires a significant effort to put the data in computer memory. I invite the authors to explain why other formats, such as NDJSON, HDF5, or Parquet, were not used.

**Ethics:**

The submitted version of the paper misses the checklist after references, but the checklist has been provided in the README file. I do not see any ethical concerns in the described study.

**Relation To Prior Work:**

Since the authors have produced EFO-1-QA using depth-first search, it will be useful to compare it to similar datasets, showing the similar and dissimilar parts.

**Summary And Contributions:**

This paper presents EFO-1-QA, a new synthetic dataset to benchmark the combinatorial generalizability of complex question answering (CQA) models. It provides a detailed description of the combinatorial generalizability problem and an overview of the proposed dataset but has a weak positioning. I recommend the authors change the paper's focus from the synthetic dataset to the comprehensive and systematic benchmark of useful CQA models.

---

> ### Author Response · Authors · 2021-09-24
> **Response to Reviewer 7kZd**
>
> Dear Reviewer,
>
> Thanks for your comments about the organization of this paper. We slightly re-organized our presentation in order to highlight our findings and the comparisons with previous datasets. Besides, we released the Github repository with an MIT license. The OneDrive shareable link to the data we used (near 5 GB) is also provided.
>
> >  The evidence that real-life CGA on KGs benefits from tuning or evaluating combinatorial generalizability datasets.
>
> First of all, we want to emphasize that the combinatorial nature of queries is ubiquitous. In real world applications such as multi-turn dialogue/QA, the answer of the last query can trigger a new query as one of the inputs. In this scenario, queries are composed with logical structures and the possible outcomes are as combinatorial as discussed in this paper. When grounded to the knowledge graph, it results in combinatorial complex queries. Thus the combinatorial generalizability of CQA models studied in this paper are very important.
>
> Moreover, the combinatorial space studied in this paper is very practical. Actually, the example query in Figure 1, though very easy to understand, has the 3 anchor nodes and max projection/negation chains of length 3. It is already the most complex query studied in our paper. Thus, the types discussed in this paper are closely related to our everyday application.
>
> >  I recommend the authors change the paper's focus from the synthetic dataset to the comprehensive and systematic benchmark of useful CQA models.
>
> Following your suggestions, we update the structure of this paper to reach a balance between faithful technical details and insightful findings. Those findings have already stated in the previous version of our paper. We highlighted them to make it more accessible. We want to invite you to go through SECTION 4 for in-depth comparisons with previous datasets and SECTION 6 for highlighted findings that are only derived from our EFO-1-QA benchmark.
>
> By comparing to BetaE and other existing dataset, EFO-1-QA benchmark is far more comprehensive, generally harder and fairer. (SECTION 4)
>
> By evaluating the EFO-1-QA benchmark (SECTION 4 and 6), four highlighted findings are
> - Finding 1: Negation queries ares not significantly harder.
> - Finding 2: More training query types do not necessarily lead to better performance.
> - Finding 3: More complex queries do not necessarily have to worse performance.
> - Finding 4: There is no rule of thumb for choosing the best normal form.
>
> We note that finding 2 and 4 reveals two open problems about how to train and inference the CQA models. Finding 1 and 2 are also non-trivial and can only observed in our evaluation.
>
> Best regards,
>
> Paper206 Authors

---

> ### Author Response · Authors · 2021-09-24
> **A note for dataset storage on disk and usage in memory**
>
> This note is intended to provide some background knowledge related to reviewer's concerning about CSV format.
>
> > I am concerned about the convenience of the current dataset representation in CSV, as it requires a significant effort to put the data in computer memory. I invite the authors to explain why other formats, such as NDJSON, HDF5, or Parquet, were not used.
>
> The advantage of CSV is that one can see the dataset easily. One can open CSV  with widely used Microsoft Offices, LibreOffice and other GUI softwares. Besides, the CSV format is also widely used to release data in Kaggle.com.
>
> Next, We discuss the convenience of dataset presentation in three aspects.
>
> ### 1. Reasonable size that can be easily shared.
> The compressed zip files are no more than 5GB in total, please check the shared LINK. Given that the zip version of BetaE dataset is about 500MB, and EFO-1-QA is 20 times larger than BetaE, the 5GB storage on the disk is efficient and fair.
>
> ### 2. Comment on the mentioned format as invited
> 1. Given the space cost in DISK is fair for the compressed version, what really matters is the space cost in the MEMORY. We note that the space cost in the MEMORY is essentially affected by the IN-MEMORY data structures rather than the serialization format on the DISK, such as CSV, NDJSON, HDF5 or Parquet. In this way, the concern about MEMORY usage is not very related to the specific storage format. Instead, no matter what format one uses in the DISK storage, we use the parsed OpsTree in `fol/foq_v2.py` to store the query type information with the grounded relations and entities.
> 2. NDJSON also stores the information in texts. Moreover, NDJSON stores the field name and the field data in each line, so it contains more redundant information than CSV and is generally larger than CSV format on the DISK.
> 3. HDF5 and Parquet are more compact for serializing the data in the DISK. But after loading the same number of samples into the MEMORY, what actually matters is the data structure used, rather than the format on the DISK. Meanwhile, compressed serialization makes the data less accessible then text data.
>
> ### 3. Data Management Practice
> 1. Our data organization allows more efficient dataset management practice. We don’t pack or compress everything into a single file. In fact, it is not wise to load everything into the memory unless the data must be very frequently visited. In this way, we only need to talk about how to manage the single CSV file.
> 2. We store the query data into different files according to their types, so users can choose the type of their interest freely. When evaluating on EFO-1-QA benchmark in our practice, we evaluate a single query type at a time, so the only one CSV file will be loaded at a time. For your reference, most files are no more than 50 MB. It is very light weighted. As for how to load the csv file into the memory, we suggest Python users to employ the pandas lib,
> ```python
> import pandas as pd
> df = pd.read_csv("the_csv_file_name.csv")
> ```
> Moreover, some of the structured information are managed by the EFO-1 formula, or JSON string. For the EFO-1 formula, we suggest users to employ our parser. For JSON string, one can use the Python Standard lib, also in two lines of code
> ```python
> import json
> some_dict = json.loads("the_target_json_string")
> ```
> or directly use our provided pipeline.
> 3. In modern dataset management systems such as RDBMS or Hadoop, CSV format is close to the concept of table in relational database or column storage. One can easily insert our data into any (distributed) RDBMS. Meanwhile, every query type and sample has its ID. So one can easily look up the specific record in the RDBMS by the SQL queries.

---

### Official Review · Reviewer_qTn1 · 2021-09-19
**A great dataset about evaluating the combinatorial generalizability of CQA models.**

**Rating:** 6
**Confidence:** 3
**Correctness:** The claims made in this paper is corr…

**Strengths:**

a. The larger query types and more data make the evaluation of the combinatorial generalization more comprehensive and convincing.

b. The deep analysis of normal forms and the investigation about how training query types affect the generalization are of great help to the future research.

c. The proposed method is general enough to be compatible with different choices of 143 operators.


**Weaknesses:**

Some technical  details of baselines are missing, which may make some readers confusing.

**Additional Feedback:**

No

**Clarity:**

The author clearly demonstrates the construction of the dataset and the figures and tables in the paper are vivid.

**Documentation:**

This paper provided the code to reproduce the benchmark and described the data collection and organization in detail. However, I didn't see the license?

**Ethics:**

This paper has no ethics issues.

**Relation To Prior Work:**

The proposed dataset extended the query types in previous dataset.

**Summary And Contributions:**

This paper builds  a new dataset EFO-1-QA to benchmark the combinatorial generalizability of CQA models. This dataset consists about 301 query types which is larger than previous work. Meanwhile, the author explore the impact of the different operators and normal forms.  The proposed dataset is very important and will have a great impact to the community and promote future researches.

---

> ### Author Response · Authors · 2021-09-24
> **Response to Reviewer qTn1**
>
> Dear reviewer,
>
> Thanks for your kind advices and reminder. We released the code on Github with an MIT license and our data is also shared in the OneDrive.
>
> > Some technical details of baselines are missing, which may make some readers confusing.
>
> Follow your suggestions, we add a new SECTION 4 to compare the EFO-1-QA benchmark with potential baselines (other existing datasets). Our dataset is shown to be more comprehensive for more query types, supported operators and normal forms. Notably, the comparison of EPFO queries and Negation queries on EFO-1-QA benchmark and BetaE dataset also shows that our dataset is generally harder, and fairer (not biased to a few hand-crafted query types).
>
> We also invite you to go through SECTION 4 and 6 where we have highlighted four non-trivial findings that are discovered by our pipeline. I hope these insights will help the community to understand more about CQA tasks on KG.
>
> Best Regards,
>
> Paper206 Authors

---

### Official Review · Reviewer_p6yk · 2021-09-20
**A good benchmark for combinatorial generalizability of complex query answering on knowledge graphs**

**Rating:** 6
**Confidence:** 3
**Correctness:** The method is sound and correct.
**Clarity:** This paper is well written.

**Strengths:**

1. An extendable framework is built by extending the scope from hand-crafted query types to the family of EFO-1.
2. A large-scale dataset is built for combinatorial queries.
3. New findings are observed for normal forms, training, and generalization.

**Weaknesses:**

Related work should be introduced and compared.

**Additional Feedback:**

I am not quite familiar with this research topic, but I think this paper is well written.

**Documentation:**

Some samples of the dataset are provided in the supplementary material.

**Ethics:**

No ethics problem.

**Relation To Prior Work:**

Related work should be introduced and compared.

**Summary And Contributions:**

This paper extends the scope from a few hand-crafted query types to the family of Existential First-Order queries with Single Free Variables, with an extendable framework. Based on this, a large-scale dataset of combinatorial queries is built.

---

> ### Author Response · Authors · 2021-09-24
> **Response to Reviewer p6yk**
>
> Dear reviewer,
>
> Thanks for your suggestions on the related works. We formally release our code in the Github with an MIT license, the data used for the evaluation (near 5 GB) is also released by the OneDrive shareable link. We hope our code and data can help the community understand more about the CQA task on KG.
>
> > Related work should be introduced and compared.
>
> Following your suggestions, we add an additional SECTION 4 in the paper to present the detailed comparison with previous datasets. We compare to three datasets in previous works with their supported query types, operators and normal forms. Specifically, we also compare the evaluation results of EFO-1-QA and the BetaE dataset. The facts revealed in the SECTION 4 show that our EFO-1-QA benchmark is more comprehensive, (in query types, supported normal forms and operators), generally harder and fairer than a few hand crafted queries.
>
> We also want to point out that previous datasets are not targeting at the combinatorial generalizability of CQA models. For example, new query types are introduced to show that their models are able to handle specific new operators such as negation. Their goal is not to evaluate the combinatorial generalizability, not to mention how to choose the normal forms and operators. Thus, the EFO-1-QA benchmark is the first one to formally justify the combinatorial generalizability, and the potential impact from operators and normal forms **from the dataset perspective**. And we do find some new unique insights from the evaluation, please check the SECTION 4 and 6 for the unique findings.
>
> Moreover, the EFO-1-QA benchmark is complex with respect to the combinatorial query types, but is still of practical significance. We restrict the depth and the width of the OpsTree to be no more than 3. An example of 3 depth and 3 width is given in Figure 1, which is still very easy to understand for humans.
>
>
> Best regards,
>
> Paper206 authors

---

### Official Review · Reviewer_SVnp · 2021-09-22
**A formal grammar for complex query answering on knowledge graphs**

**Rating:** 6
**Confidence:** 2
**Correctness:** The paper seems correct.
**Clarity:** Please see above note about explainin…

**Strengths:**

* The framework is formal, extensible, and the particular choice of operators seems to have been thought through.
* Evaluation methods follow existing works

**Weaknesses:**

This paper may not be accessible to those without a background in formal logic, which may encompass most of those in ML (including myself). This may hinder users' understanding and also adoption of this benchmark. This paper may benefit from just a bit more explanation of tools used (Skolemization, prenex normal forms, etc), and why those choices make sense for this dataset


**Additional Feedback:**

n/a

**Documentation:**

Documentation seems reasonable

**Relation To Prior Work:**

The paper might benefit from a bit more explanation of the deficiencies of previous datasets, e.g. expanding or defending the statement that "existing datasets only contain queries from very few types, which might be insufficient for the investigation of combinatorial generalization"

**Summary And Contributions:**

This paper provides a formal grammar for describing complex queries on knowledge graphs, and extends the total number of possible queries.

---

> ### Author Response · Authors · 2021-09-24
> **Response to Reviewer SVnp**
>
> Dear reviewer,
>
> Thanks for your constructive suggestions. We released our code with an MIT license on Github, and prepared our data with a shareable OneDrive link. We also tried to make our paper more self-contained and insightful following your suggestions
>
> > This paper may not be accessible to those without a background in formal logic
>
> We provide some self-contained materials for the formal definition and derivation of the EFO-1 query families from first order queries in APPENDIX A. Instead of jumping into the definitions directly like previous works, our derivation from the first order queries provided
>
> (1) the formal procedure that guarantees the desired properties of all types in the EFO-1 family.
>
> (2) the gap between the current largest EFO-1 family and the first-order query family from the perspective of formal logic.
> We emphasize that this gap clearly presents the limitation of EFO-1 queries and will hopefully encourage researchers to explore more in the direction of the general first order queries.
>
> > The paper might benefit from a bit more explanation of the deficiencies of previous datasets, e.g. expanding or defending the statement that "existing datasets only contain queries from very few types, which might be insufficient for the investigation of combinatorial generalization"
>
> Moreover, we also include an additional SECTION 4 to compare our benchmark to the previous datasets. Our benchmark contains more query types and supports more normal forms and operators. In addition, EFO-1-QA benchmark is shown to be generally harder and fairer than BetaE dataset. We believe that our large scale benchmark is important for better understanding for the complex query answering tasks. Notably, we have highlighted four non-trivial findings from SECTION 4 to SECTION 6. Those findings are only discovered by evaluations on the EFO-1-QA benchmark. Some of them refute the observations from the previous dataset. Two findings even lead to two open problems about how to train and inference the CQA models.
>
> We believe that this work, including the theoretical justification from the formal logic, code repository, benchmark data and new insights will help researchers explore deeper understanding of the CQA task on knowledge graphs.
>
> Best regards,
>
> Paper206 Authors

---

### Author Response · Authors · 2021-09-24
**Summary for updates in the rebuttal period for Reviewers**

Dear reviewers,

Thanks for your suggestions and feedbacks regarding the code, representation and analysis. We respond to each reviewer and we highlight the modifications here for your reference. All modifications are also reflected in either the submitted PDF and the Github repository.

1. We release our code on Github with the MIT license. The URL is updated into the following link
https://github.com/HKUST-KnowComp/EFO-1-QA-benchmark
2. We release the data with the OneDrive sharepoint service. The link can also be found in the README.md in the Github repository
https://hkustconnect-my.sharepoint.com/:f:/g/personal/zwanggc_connect_ust_hk/EpaFL1PUoOFBuCc7hclIM30B8c21e-Tnv1gL11jw_z_SQQ?e=m8RJb5
3. We add a note in the APPENDIX A to introduce some self contained background of formal logic, including the definitions of prenex normal form and the formal definition of Skolemization.
4. We add in-depth data analysis and the comparison with previous datasets in SECTION 4 and 6 to distinguish our generated benchmark from the existing works. Our analysis shows that
    1. Our dataset is more comprehensive, fairer and harder than previous datasets, see Table 4 and 5 in SECTION 4.
    2. Our evaluations reveal non-trivial conclusions and refute some conclusions drawn from previous datasets.

> - Finding 1: Negation queries ares not significantly harder.
> - Finding 2: More training query types do not necessarily lead to better performance.
> - Finding 3: More complex queries do not necessarily have to worse performance.
> - Finding 4: There is no rule of thumb for choosing the best normal form.

In short, the analysis on our dataset shows that our dataset is of higher quality and reveals non-trivial conclusions that can not be drawn from any previous datasets. We hope this work can help future research.

Best regards,

Paper206 Authors

---

### Decision · Program_Chairs · 2021-10-09

**Decision:**

Accept

**Comment:**

This paper presents a dataset for complex question answering. The dataset is significantly larger than previous datasets, and this problem space is underexplored but important. Reviewers' comments were addressed well in the rebuttals. All reviewers agree that this paper has merit and potential contribution for future research in this topic.